# Bona Fide Tumor Suppressor Genes Hypermethylated in Melanoma: A Narrative Review

**DOI:** 10.3390/ijms221910674

**Published:** 2021-10-01

**Authors:** Canan Güvenç, Fien Neckebroeck, Asier Antoranz, Marjan Garmyn, Joost van den Oord, Francesca Maria Bosisio

**Affiliations:** 1Department of Dermatology, University Hospitals Leuven, 3000 Leuven, Belgium; fien.neckebroeck@student.kuleuven.be (F.N.); marjan.garmyn@uzleuven.be (M.G.); 2Department of Imaging and Pathology, Translational Cell and Tissue Research, University Hospitals Leuven, 3000 Leuven, Belgium; asier.antoranzmartinez@kuleuven.be (A.A.); joost.vandenoord@kuleuven.be (J.v.d.O.)

**Keywords:** tumor suppressor genes, epigenetic, hypermethylation

## Abstract

Loss-of-function events in tumor suppressor genes (TSGs) contribute to the development and progression of cutaneous malignant melanoma (CMM). Epigenetic alterations are the major mechanisms of TSG inactivation, in particular, silencing by promoter CpG-island hypermethylation. TSGs are valuable tools in diagnosis and prognosis and, possibly, in future targeted therapy. The aim of this narrative review is to outline bona fide TSGs affected by promoter CpG-island hypermethylation and their functional role in the progression of CMM. We conducted a systematic literature review to identify studies providing evidence of bona fide TSGs by cell line or animal experiments. We performed a broad first search and a gene-specific second search, supplemented by reference checking. We included studies describing bona fide TSGs in CMM with promoter CpG-island hypermethylation in which inactivating mechanisms were reported. We extracted data about protein role, pathway, experiments conducted to meet the bona fide criteria and hallmarks of cancer acquired by TSG inactivation. A total of 24 studies were included, describing 24 bona fide TSGs silenced by promoter CpG-island hypermethylation in CMM. Their effect on cell proliferation, apoptosis, growth, senescence, angiogenesis, migration, invasion or metastasis is also described. These data give further insight into the role of TSGs in the progression of CMM.

## 1. Introduction

Cutaneous malignant melanoma (CMM) is the most aggressive form of skin cancer. If untreated, melanomas have a serious metastatic potential and cause a significant mortality [1]. The worldwide incidence is rising, and survival is dependent on early detection [2]. CMM arises from melanocytes, i.e., melanin-producing cells that are found in the stratum basale of the epidermis. Melanomagenesis is a stepwise process in which melanocytes undergo a series of transformations. Genetic and epigenetic changes in the melanocytes, together with changes in the skin microenvironment, contribute to the transition toward malignancy [3]. 

Genes important in the development and progression of cancer can be divided into oncogenes and tumor suppressor genes (TSGs). Whereas the former become active through gain-of-function events in proto-oncogenes, TSGs, on the other hand, normally encode proteins that prevent cancer development. They function mainly through the negative regulation of cell cycle progression, stimulation of DNA damage repair, induction of apoptosis, inhibition of metastasis (for example, by cell adhesion regulation) or gene transcription regulation [4]. Their contribution to development or progression of cancer occurs through loss-of-function events [5]. In melanoma, both oncogenes and TSGs are important subjects of investigation. Multiple research articles have found evidence for the importance of several TSGs in melanoma development or progression [6,7,8,9,10]. Well-known TSGs inactivated in melanoma are PTEN, P53 and the familial CDKN2A [11,12,13]. In addition, many other aberrant genes in CMM have been suspected to serve as TSGs, but firm evidence is often lacking. Therefore, “bona fide” TSGs are distinguished from candidate TSGs. According to DaCosta et al., bona fide TSGs meet at least one of the following criteria: (a) in vivo malignancy is suppressed when wild-type gene is present in cell lines mutant for the gene; (b) a knock-out animal for the gene is prone to develop tumors; and (c) individuals with a cancer predisposition syndrome have germ-line inactivating mutations for the genes that are not present in normal family members [14]. For the inclusion in our review, we used modified DaCosta criteria (see methods section). By contrast, we considered candidate TSGs as genes with evidenced loss of expression in melanoma samples and cell lines, but without evidence for the functional role of the gene loss in melanoma development. Therefore, in many studies searching for bona fide TSGs [15,16,17], the focus lies on the functional role of the inactivated or “silenced” TSG in the development and progression of cancer. In animal studies, siRNA and CRISPR/Cas can be used to silence candidate TSGs in order to find evidence to call them bona fide TSGs [18,19]. Multiple mechanisms may lead to the inactivation of TSGs. First, cellular mechanisms are the least frequent and include ubiquitin-proteasomal degradation, cellular mislocalization and aberrant transcription factor regulation, all resulting in very low quantities or complete lack of the tumor suppressor protein [20]. Second, genetic modifications result in changes of the DNA sequence, also referred to as mutations. These modifications can consist of chromosomal changes, resulting in entire gene deletion, or smaller point mutations, leading to loss of function of the gene [5]. Third, epigenetic modifications (i.e., DNA methylation, histone modification, chromatin remodeling and non-coding RNAs such as miRNA) do not change the DNA sequence, but do change the expression of the gene [1,21]. Their effect lies in the prevention of transcription of the gene or in the inhibition or removal of the transcribed mRNA.

In recent years, the importance of the most common form of epigenetics namely DNA methylation in the biology of CM has been increasingly acknowledged. Moreover, DNA methylations are more easily reversible than genetic variation, and therefore offer a significant opportunity for cancer therapy. As DNA methylation is extremely interesting for therapeutic purposes, an overview of the bona fide TSGs methylated in melanomas is lacking in the literature. DNA methylation in mammals generally means the addition of a methyl group to the 5-carbon position of a cytosine in CpG sites by DNA-methyltransferases [1,22,23]. Normally, CpG islands are largely unmethylated and are preferentially located in gene promoters [22,23]. DNA methylation aberrations are proven to be prevalent in melanoma; global hypomethylation and focal, gene-specific hypermethylation have been found in the melanoma cell genome. DNA promoter hypermethylation results in transcriptional silencing of TSGs [22,23,24]. By contrast, hypermethylation of gene bodies is associated with oncogene upregulation [21,22]. DNA methylation can serve as a molecular biomarker for the diagnosis and prognosis of melanoma and for the prediction of treatment response [1,23,24]. As body fluids such as blood samples can be used to analyze TSG promoter methylation, non-invasive detection is possible [23]. DNA hypermethylation is reversible in nature and can thus serve as a useful therapeutic target by demethylating agents compared to the genetic alteration (such as deletion and mutations, which are more challenging to correct) [24,25]. Unfortunately, these agents act on the whole genome, and gene-specific demethylation is still not possible. Therefore, a better understanding of the mechanisms by which DNA methylation affects cell proliferation and differentiation and melanoma progression will facilitate the development of therapeutic strategies. In this narrative review, we aim to provide an overview of tumor suppressor genes affected by epigenetic (i.e., promoter CpG-island hypermethylation) events in melanoma and to clarify what is already known about the functional role of these genes in melanoma development or progression.

## 2. Results

Overall, we included 23 studies in this narrative review using systematic search (Figure 1). We found 24 bona fide TSGs methylated in melanomas. Eight bona fide TSGs that can be methylated as well as deleted in melanomas were also found, but the discussion of genetic alterations is beyond the scope of this article (Appendix A). We extracted data about the role of the encoded proteins, the involved pathway, the expression of the gene and mechanism of silencing in melanomas (see text). The characteristics of each study are summarized in Table 1.

### 2.1. WFDC1

WFDC1 encodes a protease inhibitor involved in growth regulation and apoptosis. Liu et al. found WFDC1 expression to have reduced or lost 80% of the examined melanoma cell lines and tissues. Promoter hypermethylation contributes to WFCD1 gene silencing [26].

Re-expression of the gene in cell lines lacking the gene caused no significant changes in proliferation, colony formation or invasion. A significant increase in migration was seen in one cell line, but not in the other (highly migratory) cell line. By contrast, in vivo assays showed significant tumor growth inhibition when transfecting the above cell lines in mice. The effect is possibly related to the upregulation of DKK-1, which is a Wnt-signaling pathway inhibitor [26].

RASSF1A, RASSF8 and RASSF6 encodes a protein of the Ras-association domain family. RASSF1A encodes a microtubule-associated protein involved in the regulation of cell proliferation, migration and apoptosis. Immunohistochemistry showed decreased RASSF1A expression in melanoma cells compared with normal melanocytes or benign lesions. A reverse correlation between RASSF1A staining intensity and lymph node metastasis was observed [27]. The gene caused decreased viability, G1-arrest and apoptosis in melanoma cells. Subcutaneous injection of the RASSF1A cells versus the control wild-type (WT)-cells in nude mice resulted in significantly smaller xenograft tumors in the RASSF1A group due to decreased cell proliferation and increased apoptosis [27]. RASSF8 are important in biological processes such as cell death, cell cycle control, microtubule stability, promoter methylation, vesicle trafficking and response to hypoxia. RASSF8 expression was found significantly lower in metastatic melanoma cell lines compared with primary melanoma/melanocyte cell lines [6]. The same inverse correlation was seen in melanoma tissues: RASSF8 expression was lower in stage III/IV advanced melanomas than in stage I/II early melanomas. Metastatic melanoma tissues had significantly higher hypermethylation levels of RASSF8 compared with earlier stage tumors. Overall survival of melanoma patients with high RASSF8 expression was better than those with low RASSF8 expression [6]. Wang et al. over-expressed RASSF8 in a melanoma cell line with low intrinsic expression, resulting in increased cell cycle arrest and apoptosis as well as decreased cell growth, colony formation, migration and invasion. By contrast, RASSF8 shRNA injection in a cell line with high intrinsic expression resulted in increased cell growth, colony formation, migration and invasion. Injection of nude mice with a melanoma cell line expressing high levels of RASSF8 or the same cell line with RASSF8 shRNA resulted in an increased tumor growth and size in the latter group [6].

RASSF6 is involved in pro-apoptotic signaling pathways. Mezzanotte et al. found RASSF6 promoter methylation in the majority of the examined primary melanomas and brain metastases [46].

Vector-induced expression of the gene in a cell line with methylated RASSF6 showed decreased cell growth and invasion. It is thought that RASSF6 works through MAPK-pathway and AKT inactivation [46].

### 2.2. SOCS1

SOCS1 encodes a protein that is important in ubiquitination and negative regulation of pro-inflammatory cytokine signal transduction. Previous studies showed the frequent silencing of SOCS1 via promoter hypermethylation in human cancer, including melanoma [28,29]. Transfection of human melanoma cell lines without or with very low levels of endogenous SOCS1 resulted in significantly decreased growth due to cell cycle arrest as well as increased apoptosis [29]. Intradermal injection of SOCS1-transfected human melanoma cell lines in nude mice resulted in fewer and smaller tumors [28]. Intratumoral injection with an adenovirus expressing SOCS1 in nude mice resulted in significantly reduced tumor volumes [29].

### 2.3. TSPY—CYBA—MTA2—MX1—RPL37A—HSPB1

Gallagher et al. studied 1 parental and 3 derivative cell lines with variable tumorigenicity, cytokine resistance and metastatic potential. A total of 44 genes were downregulated in the derivative cell lines with respect to the parental cell line, of which 13 were reactivated after 2’ -deoxy-5-azacytidine (DAC) treatment (DAC functions as a DNA methylation inhibitor). Six of these (TSPY, CYBA, MTA2, MX1, RPL37A and HSPB1) have 5’ CpG islands at their promotor site and are therefore likely to be regulated by DNA methylation [30]. This was confirmed by the fact that DAC treatment of the cell lines and, thus, reactivation of the genes downregulated by DNA methylation, resulted in the suppression of cell growth and migration [30]. Similar results were obtained in vivo after treatment of mice with DAC. Of all genes, only the re-expression of TSPY mRNA maintained after DAC withdrawal [30].

### 2.4. SYK

SYK encodes a protein that is not only involved in hematopoietic cell signaling, but also in regulating cellular senescence [31]. SYK was nearly undetectable or absent in all cell lines examined by Bailet et al. and Hoeller et al., mainly due to promoter hypermethylation [31,47]. The introduction of SYK in cell lines lacking the gene resulted in less proliferation, invasion and migration and more senescent cells in the study of Bailet et al. In vivo transfection resulted in reduced tumor size and number of metastases [47].

### 2.5. H-cadherin and E-cadherin

CHD13 encodes the adhesion molecule H-cadherin. H-cadherin is strongly reduced or lost in 70% of the melanoma cell lines examined by Kuphal et al. [32]. On immunohistochemistry, H-cadherin expression is significantly less expressed in melanoma as compared with normal melanocytes and benign nevi. Seven melanoma cell lines with silenced H-cadherin expression re-expressed H-cadherin after demethylation treatment [32]. In vitro forced over-expression of H-cadherin resulted in reduced numbers and size of colonies as well as reduced migration and invasion. Proliferation did not significantly differ. In vitro use of siRNA against CDH13 resulted in increased invasiveness. Subcutaneous injection of the above cell lines in nude mice caused reduced tumor formation rate [32]. CDH1 encodes e-cadherin, a cell–cell adhesion protein important in epithelial cell function, maintenance of tissue architecture and cancer suppression. Venza et al. found a strong reduction or loss of E-cadherin in some 40% of the examined melanomas, with about 90% of them being downregulated by hypermethylation. Correlation studies revealed a link between low E-cadherin and head-neck site, ulceration, mitotic index, lymph node and distant metastasis status and a shorter (disease-free) survival [33]. Treatment of melanoma cell lines with the demethylating agent 5-Aza-dC resulted in the reactivation of E-cadherin and a significantly decreased invasion [33].

### 2.6. TRIM16

TRIM16 encodes a protein of the TRIM family, which is involved in the regulation of cell cycle, proliferation, differentiation, ubiquitination, apoptosis, tumor suppressor functions and oncogenesis [34]. Sutton et al., in 2014, found TRIM16 to be significantly downregulated in melanoma cell lines compared with normal melanocytes. This was caused partly by methylation and partly by mutations. Gene expression reduced along with tumor progression. Low TRIM16 in metastases was associated with poor prognosis [35]. The overexpression of TRIM16 in melanoma cell lines resulted in reduced cell proliferation and migration. By contrast, the knockdown of the gene in WT melanocytes resulted in increased migration [35]. Next to an endogenous tumor-suppressing role in melanocytes, TRIM16 has also an exogenous effect through keratinocytes. Heterozygous KO mice developed more and larger melanomas and more lymph node metastases. Homozygous KO mice developed exclusively larger melanomas [34].

### 2.7. RUNX3

RUNX3 encodes a transcription and cell shape-regulating protein. Previous studies showed a significant decrease in RUNX3 expression in melanoma cell lines compared with melanocytes [48]. In melanoma tissue, the expression of RUNX3 was decreased during cancer progression and was correlated with stage and five-year survival [49]. RUNX3 was strongly methylated in a metastatic melanoma cell line, and its expression was significantly elevated after 5-Aza-CdR (demethylating agent) [37]. Kang et al. re-expressed RUNX3 in cell lines lacking the gene, causing decreased cell growth [37]. An experiment by Zhang et al. showed changes in cell shape after ectopic RUNX3 expression. This was caused by enhanced formation of stress fibers and focal adhesion complexes (by increased fibronectin expression). The migratory and invasive ability of the cells were also significantly decreased [36]. In vivo, treatment of mice bearing a RUNX3-downregulated melanoma cell line with the demethylating agent resveratrol showed significantly reduced tumor growth (size and weight) in combination with increased RUNX3 expression [37].

### 2.8. APC

APC encodes a protein that regulates cellular β-catenin, a key effector in the Wnt-signaling pathway. Additionally, it regulates cell cycle, adhesion, migration, microtubule assembly and chromosome segregation. Inactivating mutations and hypermethylation of APC have been found responsible for reduced expression in melanoma cell lines and tissue [38].

KO of APC in melanoma cell lines by Worm et al. resulted in increased proliferation (without significant Wnt-signaling upregulation). However, a large (>70%) reduction in APC levels caused a decrease in invasiveness [38]. Worm et al. suggested that a certain level of APC is necessary for invasive growth. This shows that the tumor suppressor function of melanocyte APC is dose dependent: silencing of APC can contribute to melanoma development when the expression is reduced to a level that increases cell proliferation, but does not decrease the invasiveness [38].

### 2.9. MAPK13

MAPK13 encodes a protein involved in the regulation of proliferation, differentiation, survival and migration. Gao et al. found the gene to be expressed in normal melanocytes and fibroblasts, but not in melanoma cells due to promoter methylation. This process occurred in most of the examined primary melanomas, but not in benign nevi, and even higher percentages of methylation were found in metastases [39]. MAPK13 re-expression in cell lines with hypermethylation of this gene resulted in the suppression of proliferation [39].

### 2.10. RARβ

RARβ encodes a retinoic acid receptor involved in vitamin A-mediated growth inhibition and melanocytic differentiation [50]. The gene is known to be silenced by hypermethylation in 45% of the examined melanoma cell lines [40]. Dahl et al. suggested a linkage between RARβ and p14ARF. P14ARF is one of the two proteins encoded by CDKN2A. It is involved in cell cycle regulation and senescence [51]. Its expression is decreased in melanoma compared with nevi tissues and is associated with stage, Clark grade and lymph node metastasis. CDKN2A homozygous deletions, mutations and p14ARF histone de-acetylation are described in melanoma cell lines and tissues. All-trans retinoic acid (ATRA), a ligand of RARβ, was administered to melanocytes and resulted in decreased cell proliferation and increased number of cells in G1-phase, senescent cells and expression of p14ARF. These effects were reduced in melanocytes without p14ARF and in melanoma cell lines with hypermethylated RARβ. Therefore, the growth-inhibiting effect of ATRA depends on a functional RARβ-p14ARF-pathway. Furthermore, treatment of cell lines bearing hypermethylated RARβ with a demethylating agent resulted in increased RARβ expression and decreased cell growth after administration of ATRA [40].

### 2.11. AGTR1

AGTR1 encodes an Angiotensin II receptor and is thus involved in the renin-angiotensin-aldosterone system. The protein is expressed in normal human melanocytes as well as in radial growth phase melanoma cell lines (Renziehausen et al.); however, AGTR1 is severely downregulated or undetectable in the vertical growth phase and in metastatic melanoma cell lines. The same trend was found in melanoma tissues in different stages of tumor progression. There was a strong correlation between methylation and transcriptional silencing in cell lines [41]. It has been suggested that AGTR1 methylation is a late event in melanoma development [41]. Treatment with Losartan (i.e., an AT1R blocker) or AGTR1 shRNA KO in cell lines that express AGTR1 increased proliferation in serum-free conditions. By contrast, an overexpression of AGTR1 in cell lines lacking AGTR1 expression resulted in colony suppression independent of Angiotensin II [41].

### 2.12. SERPINB5

SERPINB5 encodes maspin, a protein involved in cellular adhesion. The gene is expressed only in normal human melanocytes, but not in melanoma cell lines. All melanoma cell lines examined showed promoter methylation of the gene [42]. Denk et al. transfected maspin into a melanoma cell line resulting in unchanged proliferation and migration, but reduced invasion; the latter is either a direct effect of maspin or due to MMP-2 downregulation [42].

### 2.13. 14-3-3σ

14-3-3σ encodes stratifin, a protein interacting with molecules involved in cell cycle regulation, apoptosis, mitogenic signaling and genomic stability. Schultz et al. found a significant downregulation of its expression in melanoma metastases compared with primary tumors. 14-3-3σ was methylated in all of the examined tumor tissues, and methylation increased with tumor progression [43]. 5-Aza-CdR in cell lines with highly methylated 14-3-3σ caused cell cycle arrest, partly reversible by inducing a shRNA to knock down the gene. 14-3-3σ overexpressing cells showed reduced proliferation and migration as well as increased senescence. 14-3-3σ KO in cells resulted in increased migration and reduced senescence [43].

### 2.14. TCF21

TCF21 encodes a transcription factor involved in mesenchymal-to-epithelial transition (MET) and in the repression of epithelial-to-mesenchymal transition (EMT is therefore important in cancer cell dissemination). All metastatic melanoma tissues examined by Arab et al. showed variable promoter methylation and corresponding TCF21 gene expression silencing. There was an inverse but not significant correlation between methylation status of TCF21 and overall survival [44]. This study found evidence that TCF21 regulates the expression of KiSS-1 [44], which is believed to be a bona fide TSG [52]. An overexpression of TCF21 in a cell line resulted in decreased migration [44].

### 2.15. SPINT2

SPINT2 encodes a protein serving as proteolytic inhibitor of hepatocyte growth factor activator and thereby suppressing the HGF-MET-pathway involved in cell proliferation, dissociation, migration, angiogenesis and survival. Hwang et al. found a significantly decreased SPINT2 expression and increased methylation in metastatic melanomas compared with primary melanomas. Similar data were found in melanoma cell lines, compared with human primary melanocytes [45]. Hwang et al. subsequently evaluated the functional role of SPINT2 in melanoma cell lines. An overexpression of the gene resulted in decreased cell growth, colony size and migration. By contrast, knockdown of the gene resulted in increased cell proliferation. They additionally provided evidence for MET/AKT pathway downregulation by SPINT2 [45].

## 3. Discussion

We reviewed the literature on TSGs silenced by CpG island promoter hypermethylation in CMM. Overall, 23 studies were included describing 24 bona fide TSGs affected by promoter hypermethylation in CMM. We also found bona fide TSGs that were methylated and deleted in melanoma, but since DNA hypermethylation is reversible in nature and therefore offers a significant opportunity for cancer therapy, we decided to focus on the methylated bona fide TSG in melanoma, in this review. We excluded the following genes that were methylated and deleted in melanoma: PTEN, BRIMS1, FERMT3 (KIND3), AKAP12 (SSeCKS), CDKN2A, APAF-1, MTAP and MEN1.

TSGs normally encode proteins that prevent cancer development by regulation of cell cycle progression, DNA damage repair, induction of apoptosis, inhibition of metastasis or gene transcription regulation [4]. Loss-of-function events occur in these genes in cancer [5]. However, a candidate TSG needs to meet several in vitro and in vivo conditions in order to be named a “bona fide” TSG. For the inclusion in our review, we used modified DaCosta criteria [14]. We believe these criteria are relevant and can differentiate between bona fide and candidate TSGs, emphasizing the functional role of the TSG. We identified genes lacking all of the modified Dacosta criteria as “candidate” TSGs. We combined studies focusing on the expression of TSG, silencing mechanisms and association with clinical outcomes, on the one hand, and the experimental studies meeting our criteria for bona fide TSG, on the other. To the best of our knowledge, we are the first to provide a comprehensive overview of the literature on TSGs silenced by CpG island promoter hypermethylation in CMM, focusing on their functional role in melanoma development and/or progression.

Gallagher et al. performed transcriptomic studies and observed the downregulation of some genes in high-tumorigenic melanoma cell lines compared with low-tumorigenic melanoma cell lines, with six of them showing promoter CpG-islands. They treated cell lines and mice with a demethylating agent and observed attenuated tumor growth [30]. They did not differentiate between these six genes, and it is therefore possible that one or more genes did not actually contribute to the tumor-suppressing effect. Similarly, some other studies used demethylating agents to cause the upregulation of the specific TSG expression and function [33,37,53]. As such, agents act non-specifically; it is however possible that these demethylating agents induced the expression of other TSGs besides the gene under study. Kang et al. used resveratrol in their demethylation experiments. This treatment could indeed upregulate RUNX3 expression, causing a decreased tumor growth, but other effects of resveratrol could not be excluded [37]. Finally, several studies did not discuss the pathways involved in the tumor-suppressing effect, which makes the underlying mechanism less clear [30,37,54,55]. Furthermore, some of the TSGs were involved in the same pathway. For example, APC hypermethylation of promoter 1A may have a tumorigenic effect that is independent of the Wnt signaling pathway. In particular, inhibition of APC transcripts in melanoma cells was insufficient to increase the levels of Wnt signaling, but still led to significant increases in cell proliferation rate. WFDC1 consistently upregulated the expression of the Dkk1 gene, a key gene involved in the inhibition of the Wnt signaling pathway. In addition, we observed a link between p53 pathway and SYK but also with RASSF8 [6,47].

Sutton et al. demonstrated that TRIM16 is a marker of cell migration and metastasis and is a novel treatment target in melanoma [34]. TRIM16 protein is increased with vemurafenib treatment and is required for the drug action in melanoma cells. Moreover, the combination treatment of recombinant human IFNβ and vemurafenib may reinforce each other and promote a co-operative anticancer signal. The observation of the bona fide TSGs affected by promoter hypermethylation in CMM suggests that these molecules, e.g., TCF21, SOCS, RUNX, RASSF6, RASSFA, RARβ, MAPK13, H- and E-cadherin and AGTR, are a candidate for the design of new therapeutic strategies for human melanoma. Characterization of the DNA methylation mechanisms that initiate and promote human melanoma development may identify biomarkers that could be used for prevention, early detection, treatment and monitoring of the progression of this malignancy. Therefore, future studies to assess this missing knowledge for these bona fide TSGs are needed.

## 4. Conclusions

We provided a clear overview of bona fide TSGs methylated in melanomas, and we described their functional role in melanoma development. In the absence of a broad-based definition of bona fide TSGs, we used the following criteria: (a) overexpression of the gene in cell lines or introducing the gene in KO animals results in a decrease in cancer cell-specific traits; or (b) blocking the gene in cell lines or in animals results in an increase in cancer cell-specific traits. We included 24 bona fide TSGs methylated in melanomas, and we listed their chromosomal location, physiological role, inactivation mechanism, pathway, experiments conducted to meet the bona fide criteria and hallmark of cancer acquired by TSG inactivation. Future studies can focus on animal or cell line experiments with candidate TSGs, inactivation mechanisms of bona fide TSGs and reviewing TSGs affected by other (epi)genetic inactivation mechanisms. TSGs are valuable in diagnosis, prognosis and treatment strategies.

## 5. Materials and Methods

We conducted a narrative review to overview literature on bona fide tumor suppressor genes that are methylated in melanoma. We used the electronic databases PubMed and Embase to search for eligible studies. The search strategy consisted of five concepts combined with AND. Concept 1 was tumor suppressor genes, concept 2 was melanoma, concept 3 was metastasis, concept 4 was NOT uveal or mucosal melanoma and concept 5 was methylation. These concepts were expressed in MeSH-terms (PubMed), Emtree-terms (Embase) or free text (present in title, abstract and key words). The complete search strategy can be consulted in Appendix B.

A manual inspection of the results was carried out and is presented in Figure 1. We included studies in English concerning TSGs hypermethylated in cutaneous melanoma. Studies concerning only atypical Spitz tumors, uveal or mucosal melanoma, other tumors or cancer in general were excluded. Case reports, case series and letters to the editor were also excluded. Studies not focusing on specific TSGs, but on choice of therapy, heterogeneity of methylation or techniques for monitoring expression/methylation were excluded.

This review addresses only DNA hypermethylation in CMM because, compared to irreversible genetic variation, e.g., deletion, DNA hypermethylation is more easily reversible and offers options for cancer therapy. We subsequently split the remaining results in studies in those giving evidence of bona fide TSGs, on the one hand, and those focusing on observation of expression or methylation of the genes, on the other. We used the modified DaCosta criteria for bona fide TSGs: (a) overexpression of the gene in cell lines or introducing the gene in KO animals results in a decrease in cancer cell-specific traits; or (b) blocking the gene in cell lines or in animals results in an increase in cancer cell-specific traits. To ensure that we did not overlook important studies on bona fide TSGs, we conducted a second search. For those genes for which we could not find any evidence to be bona fide TSGs in the selected studies, we searched PubMed with the following strategy: name of the gene AND tumor suppressor gene AND melanoma. We included studies on bona fide TSGs methylated in melanoma. In addition, we included some studies by checking the references of the included articles. We extracted data of all studies in the ‘bona fide TSG’ group and summarized them in Table 1 and Appendix A. We added the updated search in the flow chart and marked the additional numbers in bold and in red for the updated search.

## Figures and Tables

**Figure 1 ijms-22-10674-f001:**
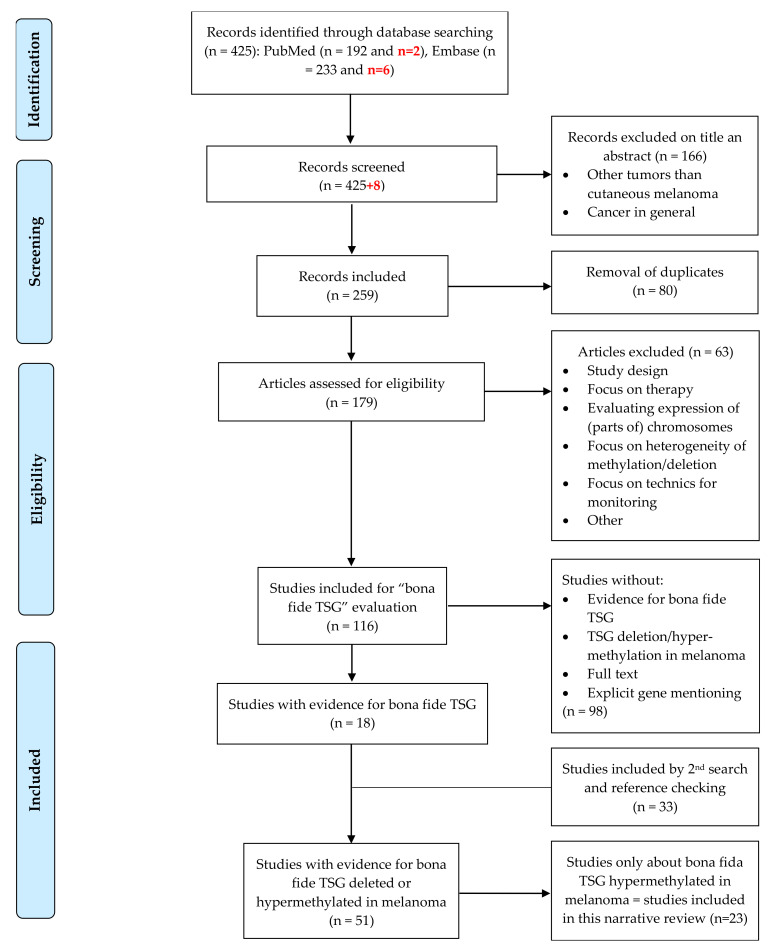
Flowchart of the study selection process. Adapted from: Moher D., Liberati A., Tetzlaff J., Altman D.G., The PRISMA Group (2009). Preferred Reporting Items for Systematic Reviews and Meta-Analyses: The PRISMA Statement. PLoS Med 6(6): e1000097. doi:10.1371/journal.pmed1000097. The marked numbers in bold and in red are the updated search (2020–2021).

**Table 1 ijms-22-10674-t001:** Bona fide tumor suppressor genes methylated in melanoma.

TSG	Protein	Location	Hallmark of Cancer	First Author	Year
WFDC1	WAP four-disulfide core domain 1	16q24.1	Tumor growth ↑	Liu S. [26]	2009
RASSF1A	Ras association domain family member 1, isoform A	3p21.31	Apoptosis ↓ Proliferation ↑ Tumor growth ↑	Yi M. [27]	2011
RASSF8	Ras association domain family member 8	12p12.1	Proliferation ↑ Migration ↑ Invasion ↑ Apoptosis ↓ Cell/tumor growth↑	Wang J. [27]	2015
RASSF6	Ras association domain family member 6	4q13.3	Growth ↑ Invasion ↑ Extravasation ↑	Mezzanotte J. [27]	2014
SOCS1	Suppressor of cytokine signaling 1	16p13.13	Cell growth ↑	Parrillas V. [28]	2012
			Cell growth ↑ Apoptosis ↓	Tagami- Nagata N. [29]	2015
TSPY	Testis-specific protein, Y-linked 1	Yp11.2	Tumor growth ↑ Migration ↑	Gallagher W. [30]	2005
CYBA	Cytochrome b-245 alpha chain	16q24.2			
MTA2	Metastasis associated 1 family member 2	11q12.3			
MX1	MX dynamin like GTPase 1	21q22.3			
RPL37A	Ribosomal protein L37a	2q35			
HSPB1	Heat shock protein family B (small) member 1	7q11.23			
SYK	Spleen associated tyrosine kinase	9q22.2	Tumor growth ↑ Invasion ↑ Metastasis ↑	Hoeller C. [30]	2005
			Tumor growth ↑ Migration ↑ Invasion ↑ Proliferation ↑ Senescence ↓	Bailet O. [31]	2009
CDH13	Cadherin 13, T-cadherin, H-cadherin	16q23.3	Migration ↑ Invasion ↑ Attachment independent growth ↑	Kuphal S. [32]	2009
CDH1	Cadherin 1, E-cadherin	16q22.1	Invasion ↑	Venza M. [33]	2016
TRIM16	Tripartite motif containing 16	17p12	Proliferation ↑ Migration ↑	Sutton S. K. [34]	2014
			Radial migration↑ Proliferation ↑ Tumor growth ↑ Metastasis ↑	Sutton S. K [35]	2019
RUNX3	Runt-related transcription factor 3	1p36.11	Cell migration ↑ Invasion ↑	Zhang X. [36]	2017
			Proliferation ↑ Tumor growth ↑	Kang S. [37]	2019
APC	APC	5q22.2	Proliferation ↑ Invasion ↓ (!)	Worm J. [38]	2004
MAPK13	Mitogen-activated protein kinase 13	6p21.31	Proliferation ↑	Gao L. [39]	2013
RARβ	Retinoic acid receptor β	3p24.2	Senescence ↓ Proliferation ↑	Dahl C. [40]	2013
AGTR1	Angiotensin II receptor type 1	3q24	Proliferation ↑	Renzie-hausen A. [41]	2019
SERPINB5	Maspin	18q21.3-q23	Invasion↑ Cell adhesion (BM-ECM) ↓	Denk A. E. [42]	2007
14-3-3σ	14-3-3*σ*, Stratifin (SFN)	1p36.11	Proliferation↑ (loss of cell cycle control) Senescence↓ Migration↑	Schultz J. [43]	2009
TCF21	Transcription factor 21	6q23.2	Mesenchymal-epithelial transition ↓	Arab K. [44]	2011
SPINT2	Serine peptidase inhibitor Kunitz type 2	19q13.2	Migration ↑ Proliferation ↑ Invasive growth↑	Hwang S. [45]	2015

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
