# Peer review of "Bona Fide Tumor Suppressor Genes Hypermethylated in Melanoma: A Narrative Review"

_ijms, 2021, doi:10.3390/ijms221910674_

Round 1
Reviewer 1 Report
In this manuscript, the authors elaborated a review on 24 genes (bona fide TSGs )silenced by promoter CpG-island hypermethylation in cutaneous malignant melanoma (CMM).
Firstly, the authors reported on the importance of this theme and the results obtained after a systematic review of the data published on diverse platforms.
The authors discussed the genes that were methylated and deleted in melanoma.
In general, the data are well corroborated and discussed, and convincingly shows that the characterization of the DNA methylation mechanisms that initiate and promote human melanoma development may lead to the identification of the biomarkers that could be used for prevention, early detection, treatment, and monitoring of the progression of this malignancy.
The manuscript is well written, concise and the appropriate references are cited.
The authors need to address the below comments to strengthen the quality of the manuscript:
Follow the template of the journal to correct the References (e.g. 12, 13, 24, 26, 49, 52).
Add reference/s to the statements from the phrase from page 9: “Also, we observed a link between p53 pathway and SYK but also with RASSF8.”.
Include a Conclusion part or A Perspective part for more information on the theme.
Check the minor spelling mistakes: fide not fida at page 8, and Fig. 1.
Reviewer 2 Report
This review Bona fide tumor suppressor genes hypermethylated in melanoma: a narrative review summarizes recent findings dealing promoter hypermethilation in melanoma. The manuscript is well-written, but some issues to fix were found.
I recommend increasing the time frame up to years 2020-2021 to improve the quality of presented review.
Second, please explain why have you exclude TERT, p16, and some other genes involved in melanoma progression. Is that gene list sufficient for the review?
Round 2
Reviewer 2 Report
You have address all the issues. Thank you.